# Extended Infusion of Beta-Lactams and Glycopeptides: A New Era in Pediatric Care? A Systematic Review and Meta-Analysis

**DOI:** 10.3390/antibiotics13020164

**Published:** 2024-02-07

**Authors:** Andrea Rahel Burch, Lukas von Arx, Barbara Hasse, Vera Neumeier

**Affiliations:** 1Department of Pharmaceutical Sciences, University of Basel, 4000 Basel, Switzerland; 2University Hospital Zurich, Hospital Pharmacy, 8006 Zurich, Switzerland; varxl@ethz.ch (L.v.A.); vera.neumeier@usz.ch (V.N.); 3Department of Chemistry and Applied Biosciences, Swiss Federal Institute of Technology (ETH Zurich), 8049 Zurich, Switzerland; 4Department of Infectious Diseases and Hospital Epidemiology, University Hospital Zurich, 8006 Zurich, Switzerland; barbara.hasse@usz.ch; 5University of Zurich, 8050 Zurich, Switzerland

**Keywords:** pediatrics, beta-lactam, glycopeptides, antibiotic, continuous infusion

## Abstract

Optimizing antibiotic therapy is imperative with rising bacterial resistance and high infection mortality. Extended infusion defined as a continuous infusion (COI) or prolonged infusion (PI) of beta-lactams and glycopeptides might improve efficacy and safety compared to their intermittent administration (IA). This study aimed to evaluate the efficacy and safety of extended infusion in pediatric patients. Adhering to Cochrane standards, we conducted a systematic review with meta-analysis investigating the efficacy and safety of COI (24 h/d) and PI (>1 h/dose) compared to IA (≤1 h/dose) of beta-lactams and glycopeptides in pediatrics. Primary outcomes included mortality, clinical success, and microbiological eradication. Five studies could be included for the outcome mortality, investigating meropenem, piperacillin/tazobactam, cefepime, or combinations of these. The pooled relative risk estimate was 0.48 (95% CI 0.26–0.89, *p* = 0.02). No significant differences between the administration modes were found for the outcomes of clinical success, microbiological eradication (beta-lactams; glycopeptides), and mortality (glycopeptides). No study reported additional safety issues, e.g., adverse drug reactions when using COI/PI vs. IA. Our findings suggest that the administration of beta-lactams by extended infusion leads to a reduction in mortality for pediatric patients.

## 1. Introduction

The global rise in antibiotic resistance due to misuse and overuse of antibiotics [1] complicates infection treatment, potentially leading to treatment failure [2,3,4,5,6], while the development of new antibiotics remains limited [7]. In pediatrics, restricted antibiotic options, with contraindications for tetracyclines and fluoroquinolones [6,8], contribute to the challenge. Approximately 37% to 61% of hospitalized pediatric patients receive antibiotics, making them the most prescribed drugs in pediatrics [9]. WHO reports that multidrug-resistant bacterial infections cause 700,000 global fatalities annually, including 200,000 newborns [6]. Optimizing antibiotic treatment is crucial for improving efficacy and safety, with rapid detection of bacterial infection and the selection of effective and safe treatments being essential for reducing mortality [10,11,12].

### 1.1. Antibiotic Resistance

The Swiss Strategy against Antibiotic Resistance (StAR) emphasizes the importance of proper antibacterial agent use to mitigate the development of antibiotic resistance and prevent infections with resistant pathogens [1].

Appropriate usage should align with pharmacokinetic (PK) and pharmacodynamic (PD) data for efficacious pathogen eradication [13]. Consequently, hospitals globally have implemented antimicrobial stewardship programs to advocate for judicious antibiotic usage [3].

### 1.2. PK/PD of Beta-Lactams and Glycopeptides

Beta-lactams and glycopeptides, the predominant antibiotics in pediatrics, are time-dependent bactericidal drugs that inhibit bacterial cell wall synthesis [14,15]. Their effectiveness depends on substantiating the free drug concentration above the minimum inhibitory concentration (MIC) for bacterial growth [15,16].

The MIC value varies depending on the pathogen and the antibiotic utilized [15]. It is crucial to maximize the fraction of time (fT) that the antibiotic concentration remains above the MIC. The minimum efficacy targets for pediatrics include fTs of >50% for penicillins, >60% for cephalosporins, and >40% for carbapenems [15]. The efficacy further increases when concentrations exceed the MIC, extending up to four times the MIC for beta-lactams [17]. For glycopeptides, the therapeutic target is an fT > MIC of 100% [15]. These objectives are frequently not achieved with the standard intravenous (i.v.) intermittent administration (IA), but they may be attainable with continuous infusion (COI) [16]. As illustrated in Figure 1, plasma concentration fluctuates with IA but remains constant with COI (own illustration). Furthermore, COI may diminish the emergence of antibiotic resistance [18], decrease the risk of inadequate antibiotic concentration, and enhance the efficacy of infection treatment [8,19].

The time-dependent pharmacokinetic/pharmacodynamic (PK/PD) mode of action of beta-lactams and glycopeptides suggests that COI might be less toxic and more effective than IA [19,20]. By minimizing peak concentrations, both COI and prolonged infusion (PI) may result in fewer adverse drug reactions (ADRs), such as nephrotoxicity, hypersensitivity reactions, or neurological deterioration [21,22].

COI also facilitates therapeutic drug monitoring (TDM) by maintaining a constant drug concentration (Figure 1B). In contrast to IA, the timing of blood sampling and antibiotic dose administration is not critical. This reduces the risk of incorrect dose adjustments [12,14]. When employing COI, the area under the curve (AUC), which is in this case used for dose adjustments, can be easily calculated by multiplying the drug concentration with the duration of application. Through COI, the attainment of the target concentration is achieved more efficiently for both glycopeptides [19] and beta-lactams [23], potentially improving clinical outcomes [24].

Evidence increasingly supports the correlation between meeting PK/PD targets and achieving clinical success in adults [12,25,26]. Although PK studies in pediatric patients suggest potential advantages of COI/PI over IA [27,28,29,30], there is limited evidence establishing its superiority in clinical efficacy and safety to warrant its adoption as the standard method for pediatric patients [8,16,31].

### 1.3. Considerations for the Pediatric Population

Developmental stages significantly alter PK (see Table 1), causing variations in antibiotic plasma concentrations based on changes in distribution volume, blood clearance, and drug half-life [31]. Primary factors contributing to this variability include body weight and maturation effects [27]. Despite demonstrated clinical benefits of COI/PI for beta-lactams and glycopeptides in adults [7,11,32], we cannot extrapolate the same outcomes for pediatric patients. SwissPedDose guidelines currently recommend IA for almost all beta-lactams and glycopeptides, with PI limited to meropenem and ceftazidime specifically for cystic fibrosis [33]. Notably, there is no recommendation for COI/PI for glycopeptides in SwissPedDose as of now [33].

### 1.4. Objective

We hypothesized that COI of beta-lactams and glycopeptides would be effective and safe in pediatrics. Our objective was to perform a systematic review with a meta-analysis to evaluate the safety and efficacy of COI/PI of beta-lactams and glycopeptides compared to IA in pediatrics, consolidating existing knowledge for a robust conclusion [35].

## 2. Results

### 2.1. Study Selection

The initial search, limited to RCTs, yielded 581 studies, with four deemed relevant for full-text screening. Following the protocol, the search was expanded to include observational studies [36]. Across EMBASE, CENTRAL, and PUBMED, we identified 3066 studies. After duplicate removal (n = 422) and title and abstract selection, we found nine studies to be included in the meta-analysis during the full-text analysis—six on beta-lactams and three on glycopeptides (Figure 2).

Figure 2 presents the results of the mentioned selection process in a flow diagram. Further exclusion details are available in Appendix A.

### 2.2. Risk of Bias Assessment

With the exception of one study, the overall risk of bias (ROB) assessment was moderate or high, as indicated in Table 2 and Table 3, with additional details provided in Appendix A.

The Wysocki study was excluded from data synthesis due to its study design, which could introduce significant bias to our outcome analysis (Appendix A). Sensitivity analysis showed no substantial modification of effect estimates after excluding studies classified as having a high ROB (Appendix A).

### 2.3. Study Characteristics

Study characteristics are outlined in Table 4 and Table 5, with additional details available in Appendix A. The administered drug doses in all studies were within the recommended range of SwissPedDose [33].

### 2.4. Data Synthesis

#### 2.4.1. Beta-Lactams: Outcome Mortality

In the COI/PI group, 2.8% (13/458) of patients died, compared to 5.6% (32/567) in the IA group [21,37,38,39,40]. The pooled RR estimate was statistically significant (RR = 0.48; CI = 0.26 to 0.89; *p* = 0.02). Except for one study, all contributing studies included the no-effect value in the CI, consistent with the prediction interval. Tau^2^ and I^2^ statistics were 0 and 0%, respectively, as Chi^2^ was smaller than the degrees of freedom (df). Visual inspection of the forest plot suggested high heterogeneity between studies (Figure 3).

Figure 3 presents the mortality outcomes associated with beta-lactam antibiotics in a forest plot. Statistically non-significant effect estimates of all outcomes and reported adverse drug reactions (ADRs) for both antibiotic groups are detailed in Appendix A.

#### 2.4.2. Subgroup Analyses

Subgroup analyses, as planned in the protocol (Appendix A), could not be conducted due to the lack of studies [36].

#### 2.4.3. Publication Bias

None of the outcomes had a sufficient number of studies (n ≥ 10) for a meaningful assessment of publication bias using funnel plots (Appendix A). As a result, no detectable publication bias was observed.

#### 2.4.4. Certainty of Evidence

We assigned a “very low” certainty of evidence for primary outcomes in both drug groups (Table 6).

## 3. Discussion

### 3.1. Overview of Findings

To our knowledge, this is the first meta-analysis comparing extended infusion with IA for glycopeptides in pediatric patients and one of the first ones for beta-lactams. Our results indicate a statistically significant lower pooled RR for mortality when administering beta-lactams via extended infusion instead of IA. This aligns with findings from adult studies, where optimal antimicrobial exposure is associated with better clinical outcomes and lower mortality [11,25]. Additionally, also very recently published data from the meta-analysis of Budai et al. align with our results [44]. Several Monte-Carlo simulation studies investigating the probability of PK/PD target attainment in pediatrics also support the use of COI for beta-lactams, further validating our findings [27,45]. Consistent with the findings of Grupper et al. [46], we did not observe COI/PI to be inferior to IA in terms of the safety and efficacy of glycopeptides. However, we did not find significant beneficial differences in mortality, clinical success, or microbiological eradication when using glycopeptides.

Potential reasons for this include the study setting, aiming for high target concentrations to combat even the most resistant pathogens. If patients had infections caused by pathogens requiring lower antibiotic concentrations, IA possibly remained sufficiently effective. Consequently, any potential superiority of COI/PI would not have been observable. Supporting this finding, clinical trials showed that clinical success was still achieved, event when most patients did not reach the target antibiotic blood concentrations [25,47]. Therefore, non-critically ill patients may not experience the same clinical benefits as critically ill patients, who typically require higher antibiotic efficacy to combat pathogens [48]. This aligns with research on COI in adults and echoes clinical suggestions applied in the clinic of Heidenheim [25,39,47,49,50]. Additionally, this assumption is supported by Shabaan et al., the only beta-lactam study included in the meta-analysis conducted on critically ill patients [21]. It was the only study to demonstrate statistically significant benefits for all three primary outcomes assessed. While direct evidence is lacking, the literature indirectly suggests that COI may reduce antibiotic resistance emergence by avoiding subtherapeutic concentrations [11,51]. However, this hypothesis requires further investigation.

### 3.2. Implications for Clinical Practice and Outlook

COI of beta-lactams and glycopeptides aligns better with the PK/PD profile than IA [49,52]. Our findings, demonstrating safety and efficacy benefits in pediatric patients, are supported by a review on COI use in the pediatric population [7,49,53,54]. Cheng et al. also reported that COI of meropenem was more effective in treating sepsis than IA [55]. COI offers additional potential benefits, such as reducing patients’ length of stay and enhancing cost-effectiveness, which can be attributed to increased therapy efficacy and align with the interests of hospitals [8,11].

We recommend conducting an RCT to further explore the comparative benefit of COI combined with therapeutic drug monitoring (TDM) versus IA for the specified beta-lactams and glycopeptides in our guideline.

### 3.3. Strengths and Limitations

Our systematic review employed well-documented methods outlined in a registered protocol, adhering to the *Cochrane Handbook of Systematic Reviews*, with a specific focus on the relatively unexplored pediatric population [36,56].

A primary limitation was the scarcity of studies available for inclusion in the meta-analysis. The identification of only six eligible beta-lactam studies (including three RCTs) and three glycopeptide studies (including one RCT) underscores the reported lack of evidence [49]. Ethical considerations and insufficient financial incentives for conducting such studies in pediatrics may contribute to the limited research. With over 50% of the studies being non-RCTs, uncontrolled confounding factors could have influenced our results. The shortage of studies constrained both publication bias assessment and subgroup analysis, necessitating the pooling of all pediatric age groups, beta-lactam drugs, COI with PI, and all disease severities. This, in turn, compromised the robustness of the pooled RR estimates. Exclusion of the Shabaan et al. study, which showed reduced mortality with COI for beta-lactams [21], would have prevented achieving statistical significance. This study had the highest weight (inverse variance), further reducing the overall robustness. However, it was the only study with a low risk of bias, enhancing the credibility of the effect estimate. To elevate the low GRADE of the findings, additional well-designed RCTs are imperative to further explore the benefits of COI.

A limitation is excluding patients on continuous renal replacement therapy (CRRT) to minimize confounding, despite our advocacy for COI for critically ill patients, many of whom require CRRT. Future research should explore CRRT’s impact on outcomes of COI of antimicrobials. Additionally, our study did not consider pharmacokinetic changes like fluid shifts or organ dysfunction associated with severe illness [57]. We were also unable to adjust for concomitant antibiotic use such as aminoglycosides, potentially confounding our findings.

### 3.4. Conclusions

Our research suggests that COI of beta-lactams and glycopeptides for pediatric patients is feasible, safe, and more efficacious. Existing PK simulation studies and those conducted with adults support the benefits of COI. However, further validation of our findings and paving the way for clinical implementation require more RCTs. Therefore, we propose conducting an RCT to investigate the comparative benefits of COI combined with TDM over IA of beta-lactams and glycopeptides, specifically in critically ill pediatric patients who are likely to derive the most benefit from COI.

## 4. Methods

### 4.1. Eligibility Criteria

Following the *Cochrane Handbook of Systematic Reviews*, we formulated a study protocol adhering to the PRISMA-P checklist to ensure transparency and comprehensiveness [56,58]. The protocol was registered in PROSPERO (CRD42023407772) before the literature search [36]. Deviations from the protocol were documented (Appendix A). We established inclusion and exclusion criteria for the publications (Table 7). For infusion durations, we defined 24 h/day as COI, ≤1 h as IA, and >1 h <24 h/day as PI.

### 4.2. Search Strategy and Information Sources

After refining the search strategy with a professional librarian [61], we systematically searched EMBASE, MEDLINE, and CENTRAL databases for relevant studies published between 1960 and 17 April 2023 (Appendix A). To filter for randomized controlled trials (RCTs), we utilized the Cochrane filter [62]. A second search, including observational studies, was conducted, as the initial search yielded fewer than 10 RCTs, as predefined in the protocol [36]. No other search filters were applied. We validated the search by checking the inclusion of defined key papers. Before data synthesis, we reran the search to include new publications. Duplicate removal was conducted manually using the deduplication tool in EndNote, comparing titles, years, and authors, followed by the digital object identifier (DOI) if available [63]. Additional relevant publications were manually sought in the references of reviews and studies included in the full-text review.

### 4.3. Study Selection

Two independent reviewers screened titles and abstracts for inclusion in the full-text assessment, with a third reviewer resolving discrepancies. Full texts of eligible studies were screened based on inclusion and exclusion criteria (Table 7). Results were compared after each step, with consensus decisions in cases of differences.

### 4.4. Data Collection and Analysis

Following the *Cochrane Handbook for Systematic Reviews*, two independent reviewers conducted the data extraction in an EXCEL table [56,64]. Applicability was tested using a sample study, and differences were resolved through consensus, involving a third reviewer when necessary. In cases of missing data, study authors were contacted via email for the required information. Relative risks (RRs) were chosen as the measure of effect for the outcomes (Appendix A). A statistically non-significant result on a *p*-level of 0.05 was indicated if the 95% confidence interval (CI) contained the value one. To prevent calculation errors due to division by zero in studies with no events in one arm, an event value of 0.5 was used, adhering to the *Cochrane Handbook for Systematic Reviews of Interventions* [56]. Studies with no events in either study arm for a specific outcome were excluded from the meta-analysis.

### 4.5. Risk of Bias Assessment

Two independent reviewers used the Cochrane tool for randomized trials ROB2 [65] to assess the risk of bias (ROB) for each outcome. Non-randomized trials were evaluated using the risk of bias in non-randomized studies of interventions (ROBINS-I) tool [66]. A sensitivity analysis was performed to gauge the robustness of the effect estimates by excluding studies with a high ROB. The reviewers reached a consensus to determine if the ROB for each study was too high for inclusion in the analysis.

### 4.6. Data Synthesis

Glycopeptides and beta-lactams data syntheses were conducted independently. A meta-analysis with a random effects model was performed for each outcome of each antibiotic group if at least two studies reported the outcome. Forest plots were generated using the meta package in R in RStudio [67,68], employing inverse variance to weight studies as per the Cochrane handbook [56]. If fewer than two studies reported the outcome, results were reported in prose.

#### 4.6.1. Subgroup Analyses and Heterogeneity Assessment

Planned subgroup analyses for both drug groups encompassed different drugs, age, sex, treatment indication, infectious agent, severity of infection, concomitant diseases, and concurrent use of other antibiotics [36]. Forest plots were utilized for qualitative heterogeneity assessment, where a significant overlap of CI indicated high heterogeneity [56]. Qualitative and quantitative heterogeneity assessment employed Chi^2^ tests, the Higgins I^2^ statistic, τ^2^, and prediction intervals calculated in R using the meta and metafor packages [56,67,69,70]. To address heterogeneity resulting from pooling PI with COI and pooling different beta-lactam antibiotics, a random effects model was chosen.

#### 4.6.2. Publication Bias

We planned to assess publication bias by visually inspecting funnel plots of primary outcomes, created using the metafor package in R [68,69]. A minimum of 10 studies is generally considered sufficient for adequate test power to assess funnel plot asymmetry [56,71]. If publication bias was detectable, we intended to use the trim and fill function to obtain an effect estimate of the true unbiased effect [56].

#### 4.6.3. Grade Assessment

We used the grading of recommendations assessment, development, and evaluation (GRADE) approach to assess the certainty of the evidence for each primary outcome [56,72].

## Figures and Tables

**Figure 1 antibiotics-13-00164-f001:**
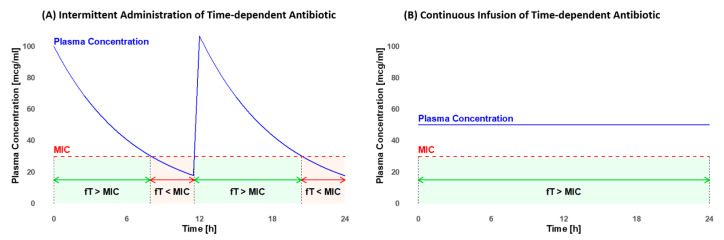
Comparison of the fraction of time (fT) that the plasma concentration remains above the minimal inhibitory concentration (MIC) in intermittent administration (**A**) and continuous infusion (**B**). In intermittent administration, there are efficacy gaps where the fT < MIC (light red). With continuous infusion, a constant fT of 100% > MIC (light green) can be maintained throughout the entire administration period.

**Figure 2 antibiotics-13-00164-f002:**
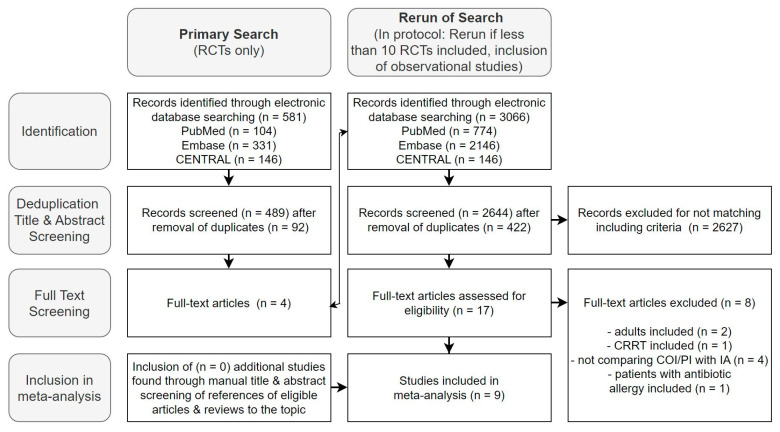
Flow diagram of the selection process of studies for inclusion in the meta-analysis.

**Figure 3 antibiotics-13-00164-f003:**
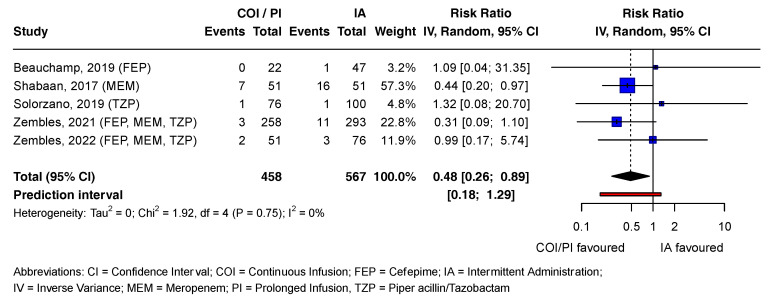
Forest plot examining mortality outcomes associated with beta-lactam antibiotics. References [21,37,38,39,40] are cited in Figure 3.

**Table 1 antibiotics-13-00164-t001:** Pharmacokinetic differences between children and adults [25,34].

PK Process	Observable Differences
Absorption	Higher gastric pH due to reduced acid production Slower gastric emptying and reduced secretion of bile salts Higher permeability of the skin
Distribution	Higher amount of body water (intracellular + extracellular) Immature blood–brain barrier
Metabolism	Less liver perfusion Less efficient metabolism due to immature liver Less efficient transport proteins Different expression of enzymes for metabolism (e.g., CYP3A7 only in newborns; CYP3A4 not yet expressed in newborns)
Excretion	Lower glomerular filtration rate Reduced tubular secretion

CYP = cytochrome P450; PK = pharmacokinetics.

**Table 2 antibiotics-13-00164-t002:** Overall risk of bias of primary outcomes in studies investigating beta-lactams.

Beta-Lactam Study	Outcome		
	Mortality	Clinical Success	Microbiological Eradication
Beauchamp 2019 [37]	High risk	High risk	High risk
Chongcharoenyanon 2021 [17]	Some concerns	High risk	n/a
Shabaan 2017 [21]	Low risk	Low risk	Low risk
Solorzano 2019 [38]	High risk	High risk	n/a
Zembles 2021 [39]	Some concerns	Some concerns	n/a
Zembles 2022 [40]	Some concerns	Some concerns	n/a

n/a not applicable.

**Table 3 antibiotics-13-00164-t003:** Overall risk of bias of primary outcomes in studies investigating glycopeptides.

Glycopeptide Study	Outcome		
	Mortality	Clinical Success	Microbiological Eradication
Demirel 2015 [41]	High risk	High risk	High risk
Gwee 2015 [42]	Some concerns	n/a	n/a
Wysocki 2022 [43]	High risk	High risk	High risk

n/a not applicable.

**Table 4 antibiotics-13-00164-t004:** Characteristics of beta-lactam studies included in the systematic review.

Study	Country/ Study Period	Study Design	AB	Sample Size (n)	Sex, f (%)	Main Indication for Treatment	Inclusion Criteria	Outcomes
Beauchamp, 2019 [37]	USA/January 2007–April 2016	CS (retrospective)	FEP	67	39	Gram-negative bacteremia	Age: 31 days to 17 years with documented Gram-negative bacteremia susceptible to FEP. (MIC ≤ 8 µg/mL.) At least 48 h of cefepime and at least 7 days of appropriate culture-directed therapy.	Mortality within 14 days of antibiotic therapy start or bacteremia relapse with the same organism, as evidenced by positive blood culture within 30 days of culture clearance, treatment failure (absence of defervescence, white blood cell normalization, and culture clearance (defervescence = drop of body temperature to less than 38.3)).
Chongcharo-enyanon, 2021 [17]	Thailand/ July 2019–April 2020	RCT (open label)	TZP	90	56	Pneumonia (32%), febrile neutropenia, and other	Age: 1 month to 18 years, body weight > 3 kg, and suspected or proven multidrug-resistant Gram-negative bacterial infection.	Piperacillin plasma concentrations mid-dosing interval.
Shabaan, 2017 [21]	Egypt/ August 2013–June 2015	RCT (open label)	MEM	102	46	Gram-negative late-onset sepsis	Age: <28 days, late-onset sepsis (sepsis after 72 h of birth), and Gram-negative bacteria sensitive to MEM. Requirement for confirmation of sepsis: positive blood, cerebrospinal fluid, urine, and/or synovial cultures.	Clinical success (complete resolution of clinical signs and symptoms of sepsis at the end of therapy (hemodynamic stability, normal arterial blood gas values, temperature stability, tolerance for enteral feeding, and discontinuation of inotropes for at least a 48 h duration). Microbiological success: eradication after 7 days of MEM therapy.
Solorzano, 2019 [38]	Mexico/April 2012–August 2015	RCT (open label)	TZP	176	41	Febrile neutropenia	Age: <18 years, haemato-oncological patients, and febrile neutropenia 8 (T ≥ 38.3 or T ≥ 38.0 over 1 h and absolute neutrophil count < 500 cells/mm^3^).	Clinical cure (no fever after 96 h of treatment or no clinical sign of infection and discharge). Clinical failure if change in treatment or death.
Zembles, 2021 [39]	USA/October 2017–March 2019	Chart analysis (retrospective)	FEP, MEM, TZP	551	41	All indications	Age: <18 years; received at least 72 h of FEP/MEM/TZP.	Length of stay, time to blood culture clearance (only patients with Gram-negative bacteremia), hospital readmission within 30 days (only patients discharged within first 14 days after completion of antibiotic therapy), and 30-day mortality after completion.
Zembles, 2022 [40]	USA/January 2013–July 2021	Chart analysis (retrospective)	FEP, MEM, TZP	124	30	Gram-negative bacteremia	Age: <18 years, confirmed Gram-negative bacteremia, and at least 72 h of FEP/MEM/TZP.	Hospital length of stay, duration of AB treatment, readmission within 30 days, all-cause mortality, time to white blood cell count normalization, time to c-reactive protein normalization, and time to blood culture clearance.

Abbreviations: AB = antibiotic; CS = cohort study; FEP = cefepime; MEM = meropenem; RCT = randomized controlled trial; TZP = piperacillin and tazobactam.

**Table 5 antibiotics-13-00164-t005:** Characteristics of glycopeptide studies included in the systematic review.

Study	Country/Study Period	Study Design	AB	Sample Size (n)	Sex , f (%)	Main Indication for Treatment	Inclusion Criteria	Outcomes
Demirel, 2015 [41]	Turkey/n/a	Chart analysis (retrospective)	VAN	77	39	Late-onset sepsis; bacteremia; other	Age: gestational age < 34 weeks, and vancomycin for late-onset sepsis.	Clinical efficacy (clinical failure), safety, and microbiological outcomes of intermittent versus continuous vancomycin infusion in preterm neonates.
Gwee, 2019 [19]	Australia/September 2014–December 2017	RCT (non-blinded)	VAN	104	50	(Suspected) sepsis; other	Age: 0–90 days old, and vancomycin administration ≥ 48 h.	Difference in the proportion of participants achieving target vancomycin levels at their first steady-state level.
Wysocki, 2022 [43]	USA/July 2010–June 2020	Chart analysis (retrospective)	VAN	28	21	Bacteremia; other	Age: >4 weeks and <18 years, and at least one serum vancomycin concentration within target range (10–20 mg/L)	Acute kidney injuries (rise in serum creatinine ≥ 1.5 × baseline, infusion reactions recorded in EMR). Treatment failure (defined as persistent positive culture for longer than or equal to 7 days, recurrence of infection within 30 days of the end of COI, or 30-day all-cause mortality).

Abbreviations: AB = antibiotic; COI = continuous infusion; EMR = electronic medical records; RCT = randomized controlled trial; VAN = vancomycin.

**Table 6 antibiotics-13-00164-t006:** GRADE evidence profile: continuous infusion of beta-lactams and glycopeptides in pediatric patients.

Study Characteristics			Quality Assessment					Number of Patients		Effect	
AB	Outcome	N, Study Type	ROB	Imprecision	Inconsistency	Indirectness	Publication Bias	COI: Outcome/ Total	IA: Outcome/ Total	RR (95% CI)	GRADE
BL	Mortality	5, nRCT	Serious	Serious	Not serious	Not serious	Undetected	13/458	32/567	0.48 (0.26–0.89)	Very low
BL	Clinical success	6, nRCT	Serious	Not serious	Not serious	Serious	Undetected	389/502	462/608	1.02 (0.87–1.19)	Very low
BL	Microbiological eradication	2, nRCT	Serious	Serious	Serious	Serious	Undetected	62/72	74/97	1.16 (0.97–1.71)	Very low
GP	Mortality	1, nRCT	Serious	Serious	n/a ^1^	Not serious	Undetected	1/35	0.5/41	2.31 (0.08–66.73)	Very low
GP	Clinical success	1, RCT	Serious	Not serious	n/a ^1^	Serious	Undetected	34/36	41/41	0.94 (0.87–1.02)	Very low
GP	Microbiological eradication	1, nRCT	Serious	Serious	n/a ^1^	Serious	Undetected	7/11	11/19	1.10 (0.61–1.98)	Very low

Abbreviations: AB = antibiotic; BL = beta-lactam; CI = confidence interval; COI = continuous infusion; IA = intermittent administration; GP = glycopeptide; GRADE = grading of recommendations assessment, development, and evaluation; N = number of studies; nRCT = not exclusively RCTs; RCT = randomized controlled trial; ROB = risk of bias; RR = relative risk (risk ratio). ^1^ Single study, inconsistency not applicable (n/a).

**Table 7 antibiotics-13-00164-t007:** Inclusion and exclusion criteria for studies for the systematic review.

Domain	Inclusion Criteria	Exclusion Criteria
Study design	RCTs; Observational studies (if less than 10 RCTs can be included); Publication language of English or German.	All other study designs; Publication language other than English or German.
Intervention	i.v. COI/PI of beta-lactam antibiotics (carbapenems, cephalosporins, monobactams, and penicillins); i.v. COI/PI of glycopeptide antibiotics (Vancomycin; Teicoplanin).	COI/PI in the context of CRRT (a potential source of bias, altering PK [59,60]).
Comparison	i.v. IA of beta-lactam antibiotics or glycopeptide antibiotics; Same treatment drug as the intervention group.	IA in the context of CRRT (a potential source of bias, altering PK [59,60]).
Outcomes (primary)	Mortality (number of events, definition according to study); Clinical success (number of events, definition according to study); Microbiological eradication (number of events, definition according to study).	Neither primary nor secondary outcomes assessed.
Outcomes (secondary)	Target attainment of PK/PD goals (qualitative assessment (quantitative assessment if possible)).	

Abbreviations: COI = continuous infusion, CRRT = continuous renal replacement therapy, IA = intermittent administration, i.v. = intravenous, PI = prolonged infusion, PK = pharmacokinetics, and RCT = randomized controlled trial.

## Data Availability

Data are available upon request from the corresponding author.

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
