# Peer review of "Extended Infusion of Beta-Lactams and Glycopeptides: A New Era in Pediatric Care? A Systematic Review and Meta-Analysis"

_antibiotics, 2024, doi:10.3390/antibiotics13020164_

Round 1
Reviewer 1 Report
Comments and Suggestions for Authors
Excellent systematic review.
Interseting, well written and a valuable contribution to the evidence base.
Would it be possible to add the PRIMA checklist to the supplementary information.

Author Response
Dear Reviewer
We thank you very much for taking the time to review this manuscript and for your valued feedback.
As suggested, we added the PRIMA checklist to the supplementary information.
kind regards,
Andrea Burch
Reviewer 2 Report
Comments and Suggestions for Authors
Dear Editors of Antibiotics Journal
I trust you are well.
Herewith kindly receive my comments regarding the manuscript entitled “Extended Infusion of Beta-Lactams and Glycopeptides: A New Era in Paediatric Care? A Systematic Review and Meta-Analysis”.
Kind regards

Minor editing and correction of grammatical errors is required.
Reviewer 3 Report
Comments and Suggestions for Authors
Thank you very much for this manuscript. The issue of antibiotic extension infusion is very important and actual problem, especially in children population. Up to day, there are not many papers concerning this issue that is why the manuscript should be published. However there is a need of further investigations in this manner. I do not have any suggestions to the authors and in my opinion the paper should be published in present form.
Author Response
Dear Reviewer
We thank you very much for taking the time to review this manuscript and for your valued feedback.
kind regards,
Andrea Burch
Reviewer 4 Report
Comments and Suggestions for Authors
This is a well-written systematic review and meta-analysis
Author Response

(The authors gave the same response as above.)
